# The Current Challenges in Developing Biological and Clinical Predictors of Congenital Cytomegalovirus Infection

**DOI:** 10.3390/ijms222413487

**Published:** 2021-12-15

**Authors:** Kenji Tanimura, Akiko Uchida, Hitomi Imafuku, Shinya Tairaku, Kazumichi Fujioka, Ichiro Morioka, Hideto Yamada

**Affiliations:** 1Department of Obstetrics and Gynecology, Kobe University Graduate School of Medicine, Kobe 650-0017, Japan; taniken@med.kobe-u.ac.jp (K.T.); akikou@med.kobe-u.ac.jp (A.U.); ima1210@med.kobe-u.ac.jp (H.I.); 2Department of Obstetrics, Hyogo Prefectural Kobe Children’s Hospital, Kobe 650-0047, Japan; sytairaku_kch@hp.pref.hyogo.jp; 3Department of Pediatrics, Kobe University Graduate School of Medicine, Kobe 650-0017, Japan; fujiokak@med.kobe-u.ac.jp; 4Department of Pediatrics and Child Health, Nihon University School of Medicine, Tokyo 173-8610, Japan; morioka.ichiro@nihon-u.ac.jp; 5Center for Recurrent Pregnancy Loss, Teine Keijinkai Hospital, Sapporo 006-0811, Japan

**Keywords:** clinical risk factor, congenital cytomegalovirus infection, neutralizing antibodies, nonprimary infection, prediction, primary infection, screening, T cell-mediated immune response

## Abstract

Congenital cytomegalovirus (CMV) infection may cause severe long-term sequelae. Recent studies have demonstrated that early antiviral therapy for infants with symptomatic congenital CMV (cCMV) infection may improve neurological outcomes; thus, accurate identification of newborns at high risk of cCMV infection may contribute to improved outcomes in affected children. However, maternal serological screening for cCMV infection by diagnosing primary infection during pregnancy, which is a popular screening strategy, is inefficient, because the number of cCMV infections with nonprimary causes, including reactivation of or reinfection with CMV, is larger than that of cCMV infections with primary causes. Low levels of neutralizing antibodies against pentameric complex and potent CMV-specific T cell-mediated immune responses are associated with an increased risk of cCMV infection. Conversely, our prospective cohort studies revealed that the presence of maternal fever/flu-like symptoms, threatened miscarriage/premature delivery, or actual premature delivery are risk factors for cCMV infection among both women with normal pregnancies and those with high-risk ones, regardless of whether the infection is primary or nonprimary. This review focused on host immune responses to human CMV and current knowledge of potential biological and clinical factors that are predictive of cCMV infection.

## 1. Introduction

Human cytomegalovirus (CMV) is the most common congenital viral infection and can lead to severe long-term neurological sequelae and even death in affected children. The prevalence of congenital CMV (cCMV) infection among newborns worldwide is estimated to be 0.7%, and 10–15% of infected fetuses have symptoms of cCMV infection at birth. The clinical manifestations of cCMV infection, including fetal growth restriction, low birth weight, and involvement of the central nervous system and multiple organs, can cause major neurological sequelae in approximately 90% of surviving affected infants. By contrast, long-term neurological sequelae, including progressive sensorineural hearing difficulty and mental retardation, can develop in 10–15% of infants with asymptomatic cCMV infection [1].

Recently, it has been reported that early antiviral therapies with oral valganciclovir may improve audiologic and developmental outcomes in newborns with symptomatic cCMV infection [2,3,4]. Particularly, a recent large-scale study demonstrated that universal neonatal screening for cCMV infection through CMV DNA polymerase chain reaction (PCR) assay in a newborn’s urine, followed by diagnostic workup and valganciclovir therapy for symptomatic cCMV infection, could reduce neurological sequelae in affected infants [4]. However, universal neonatal screening for cCMV infection is not yet popular anywhere in the world. Therefore, serological tests, including maternal blood tests of CMV-specific immunoglobulin G (IgG) and CMV-specific immunoglobulin M (IgM), are commonly used in practice because it is conventionally thought that preexisting maternal human CMV (HCMV) immunity can exert a protective effect against intrauterine transmission, and, therefore, that almost all symptomatic cCMV infections in infants are caused by maternal primary CMV infection either during or just before pregnancy. Recent studies, however, have demonstrated that preexisting maternal HCMV immunity provides only limited protection against transmission to the fetus [5], and also that the number and severity of symptoms in infants with cCMV infection from pregnant women with nonprimary CMV infection (i.e., reactivation of latent CMV or reinfection with a different strain of CMV) was similar to or higher than those in such infants from pregnant women with primary CMV infection [6,7]. In fact, the prevalence of cCMV infection in China (0.7%), where 96% of pregnant women are seropositive for CMV [8], is higher than that in Finland (0.2%), where 71% of pregnant women are seropositive [9]. These facts should be considered in the development of not only screening methods for cCMV infection but also HCMV vaccines. Additionally, to understand the mechanisms involved in the reactivation of latent CMV and reinfection with a different strain of CMV, which is involved in the origin of cCMV infection, the host immune responses to HCMV should be analyzed.

This review focuses on host immune responses to HCMV and on the current knowledge of potential biological and clinical factors that are predictive of cCMV infection.

## 2. The Struggle between HCMV and Host Immunity

### 2.1. Innate Immune Cells

Innate immune response is the first defense against HCMV infection. Myeloid cells (i.e., monocytes, macrophages, and dendritic cells) play a key role in sensing HCMV infection and producing cytokines. These cells not only possess direct antiviral activities but also play important roles in inducing adaptive immune responses [10]. However, HCMV can evade innate immune responses, for example, by impairing the antigen-presenting ability of myeloid cells by downregulation of major histocompatibility complex (MHC) proteins and costimulatory molecules. Concretely, during the early phase of viral replication, HCMV gene product U3 inhibits intracellular transport of MHC class I molecules toward the cell surface. US2 and US11 dislocate newly generated MHC class I molecules from the endoplasmic reticulum (ER) to the cytoplasm, where they are degraded by the proteasome [11].

Natural killer (NK) cells are also involved in the innate immune response to HCMV infection. NK cells can produce cytokines that stimulate other immune cells, as well as directly killing HCMV-infected cells [12]. HCMV can infect polymorphonuclear leukocytes and monocytes; conversely, however, the virus cannot infect NK cells but can affect their function. A balance of signals generated by activating and inhibitory receptors decides whether NK cells do or do not kill the infected cells. NKG2D is one of the activating NK cell receptors in humans, and its ligands include MHC class I chain-related protein A (MICA), MHC class I chain-related protein B (MICB), and UL 16-binding protein 1 (ULBP1) to ULBP6. Expression of these ligands on the infected cells induces NK cell cytotoxicity. In contrast, the UL 16 and UL 142 glycoprotein of HCMV downregulate ULBPs, MICA, and MICB by causing intracellular retention of these ligand proteins [13,14,15]. On the other hand, US 18 and US 20 downregulate MICA by targeting the ligand protein for lysosomal degradation [16]. Thus, HCMV evades NK cell attack by downregulation of the ligands of activating NK cell receptors.

Meanwhile, the cytotoxic activity of decidual NK cells is poorer than that of NK cells in the peripheral blood. However, decidual NK cells are thought to be important in preventing CMV transmission to fetuses in early pregnancy because these cells can exert cytotoxic activity when exposed to HCMV-infected decidual fibroblasts [17].

### 2.2. Adaptive Immune Response

T cells play major roles in the adaptive immune response. They recognize exogenous antigen peptides presented by MHC molecules on infected cells or by professional antigen-presenting cells. CD4^+^ T cells not only help antigen-specific B cells activate and produce antibodies but also help CD8^+^ T cells activate and exert direct cytotoxic effects against infected cells. In addition, it has been recently shown that CD4^+^ T cells exhibit antiviral activity, which is independent of their helper function, through the production of cytokines IFN-γ and TNF, and through direct cytolytic actions via perforin-dependent and Fas-dependent killing [18]. Indeed, both CD4^+^ and CD8^+^ T cells were shown to play crucial roles in the resolution of acute CMV infection in adult mouse models [19].

Previous studies have demonstrated that maternal neutralizing antibodies with high avidity to HCMV, which are produced by antigen-specific B cells activated by helper CD4^+^ T cells, can decrease the risk of cCMV infection [20].

Conversely, fetal HCMV infection was reported to induce a strong a CD8^+^ T cell response as early as 22 gestational weeks [21], and that depletion of CD8^+^ T cells led to fatal outcomes in a mouse model of cCMV infection [22]. These results demonstrated that both humoral immunity, in which CD4^+^ T cells and B cells play crucial roles, and cellular immunity, in which CD8^+^ T cells play a major role, are important in preventing the mother-to-fetus transmission of HCMV infection.

Thus, several researchers have attempted to investigate predictors of cCMV infection by measuring neutralizing antibodies (this issue is described in Section 3.2) or by evaluating the magnitude of CD4^+^ and CD8^+^ T cell responses (this issue is described in Section 4).

### 2.3. The Establishment of Latency and Reactivation of HCMV

Polymorphonuclear leukocytes and monocytes take up HCMV virus particles and express immediate early (IE) antigens. The results of previous studies lead to the hypothesis that abortively infected polymorphonuclear leukocytes and monocytes transport internalized HCMV virions into the blood to disseminate HCMV to various organs. In the bone marrow, HCMV infects hematopoietic progenitor cells and establishes latent infection in these cells. However, latent HCMV is detectable only in monocytes in the blood.

The mechanisms by which HCMV establishes latent infection in myeloid cells remain largely unknown. Some potential mechanisms for establishing latent infection are as follows: (1) viral proteins, such as UL 138 and US 28, may indirectly alter histone modification in the major immediate-early promoter (MIEP) to maintain repression of viral genes; (2) US 28 inhibits activation of c-fos and NF-κb, which activate the MIEP; (3) HCMV-encoded microRNA (miRNA) achieves transcriptional repression; (4) HCMV genome may include binding sites for myeloid-specific repressive transcription factors, such as KRAB-associated protein 1, and they recruit co-repressor complexed to suppress viral gene expression; (5) HCMV RNA contains binding sites for cellular miRNAs which suppress viral genome expression, etc.

On the other hand, the MIEP is thought to be a major regulator of HCMV latent infection and reactivation. In the cells latently infected with HCMV, the MIEP is heterochromatinized and is occupied by chromatin repressor complexes. Inflammation, DNA damage, and oxidative stress, etc. can activate the MIEP by triggering the replacement of these repressors by activating transcription factors (e.g., NF-κb, AP-1, CREB), co-activators, and histones with activating modification before latent HCMV finally achieves reactivation [23].

## 3. Prediction of cCMV Infection by Serological Assays

### 3.1. CMV-Specific Antibody Tests and CMV-Specific IgG Avidity Measurements

It is conventionally thought that the majority of symptomatic cases of cCMV disease are caused by primary infection either during or just before pregnancy [24]. Thus, serological tests for detecting primary CMV infection, including maternal blood tests of CMV-specific IgG and CMV-specific IgM, have been widely used in pregnant women. However, positive results for CMV-specific IgM are 20–25% sensitive, and the false-positive rate for detecting primary CMV infection is 15–20% because CMV-specific IgM may persist in serum for 6–9 months after primary CMV infection. Thus, the serum CMV-specific IgG avidity index is used to confirm primary CMV infection [25]. Because the CMV-specific IgG avidity index increases over time, a low CMV-specific IgG avidity index, which indicates a recent CMV infection, was reported to be a significant predictor of cCMV infection [26].

Indeed, 40% of fetuses whose mothers have primary CMV infection during pregnancy will have cCMV infection. Nevertheless, preexisting maternal HCMV immunity provides only limited protection against intrauterine HCMV transmission [5]. Furthermore, the majority of cases of cCMV disease are caused by nonprimary maternal CMV infection [6,7]; for example, 75% of cases of cCMV disease that occur annually in the United States are caused by nonprimary maternal infection [27]. Thus, the utility of serological tests in predicting cCMV infection is limited.

### 3.2. Epitope-Specific Antibody Detection

The major targets of HCMV-specific antibodies are viral glycoproteins on the surface of the virions, including gB, the gM/gN complex, the gH/gL complex, and a pentameric complex, gH/gL/UL128/UL130/UL131A. Particularly, antibodies against the pentameric complex possess potent neutralizing activities. A delay in the production of maternal antibodies against the pentameric complex during primary infection is associated with an increased risk of cCMV infection [28]. Conversely, both antibodies against the pentameric complex and a higher CMV-specific IgG avidity index are correlated with decreased risk of cCMV infection [29], as described in Section 2.2. Additionally, competitive enzyme-linked immunosorbent (ELISA) assay (inhibition of monoclonal antibody binding assay) helped detect neutralizing antibodies in 10 pentameric complex epitopes (sites 1–10), which showed that an early potent antibody response to antigenic site 7 in the pentameric complex was associated with a decreased risk of cCMV infection [28]. Therefore, the pentameric complex is a major target of vaccines being developed to prevent cCMV infection, and the measurements of antibodies against the pentameric complex may improve risk assessment for cCMV infection.

Conversely, the results of previous studies suggested that the fetuses of pregnant women who were reinfected with different strains of HCMV were at a higher risk of cCMV infection [30]. To identify reinfection with different strains of CMV, Novak et al. developed an ELISA method that was based on defined heterogeneity in the antibody binding epitopes on envelopes gB and gH of laboratory (AD169 and Towne) strains of CMV [31]. They used this ELISA assay to measure serum levels of strain-specific antibodies in 96 seropositive women and found that 58 (60%) were positive for at least one of the antibodies against the four antigens, while 18 (19%) were positive for two or more antibodies. These results indicated that this ELISA method may be useful for identifying reinfection with different strains. However, this assay could not identify reinfection with any strains of HCMV except for the AD169 and Towne strains, or with the two strains that had different polymorphic epitopes on gB and gH.

Additionally, these assays for detecting epitope-specific antibodies are not commonly available because they are nonstandardized in-house assays, and data interpretation is complex.

## 4. Assays for Measuring CMV-Specific T Cell-Mediated Immunity

The results of previous studies have suggested that CMV-specific T cell-mediated immune responses play a crucial role in controlling viral replication and the severity of CMV disease. Interferon-γ (IFN-γ) release assays, including the enzyme-linked immunosorbent spot (ELISpot) assay and the QuantiFERON assays (QIAGEN, Hilden, Germany), are commonly used to evaluate T cell-mediated immunity. In the CMV ELISpot assay, the amount of IFN-γ secreted by both CD4^+^ and CD8^+^ T cells is measured after these cells are stimulated with a mixture of peptides derived from the CMV antigens IE-1 and pp65 in purified peripheral blood lymphocytes. In the CMV QuantiFERON assay, the amount of IFN-γ secreted is measured after stimulation of CD8^+^ T cells with a cocktail of peptides from various CMV proteins that bind to a range of different human leukocyte antigen class I haplotypes in whole blood.

Because strong cellular immunity can reduce the severity of CMV disease as described in Section 2.2, the CMV ELISpot assay is used to predict the risk of reactivation of CMV infection or infection with CMV in recipients of hematopoietic cell transplants [32]. Conversely, Saldan et al. measured the CMV IgG avidity index and performed CMV ELISpot assays in 80 pregnant women, including 57 with primary and 23 with nonprimary CMV infection [33]. Contrary to expectations, they found that higher CMV ELISpot levels were associated with an increased risk of cCMV infection in the fetus, especially among pregnant women with CMV IgG avidity indexes of <25% (i.e., pregnant women with primary CMV infection). For example, among pregnant women, when the CMV IgG avidity index was <25% and the CMV ELISpot result was 445 spots per 2 × 10^5^ mononuclear cells in peripheral blood, the prevalence of cCMV infection reached 50% [33]. The authors hypothesized that strong T cell-mediated immune responses might induce proinflammatory status in the placenta, and that such a condition might induce the expression of atypical molecules or receptors that enhance the passage of the virus across the placenta. Additionally, Forner et al. found that CMV ELISpot levels and maternal viremia and viruria were positively associated with the incidence of cCMV infection, but CMV QuantiFERON levels were not [34]. They speculated that an altered CMV-specific CD4^+^ T cell response that was revealed by the CMV ELISpot assay but not by the CMV QuantiFERON assay might promote the occurrence of cCMV infection [34]. However, interpreting the results of these assays is sometimes difficult.

## 5. Clinical Factors Associated with the Occurrence of Congenital CMV Infection

Recent studies have demonstrated that the number and severity of symptoms in congenitally infected infants of women with nonprimary CMV infection were similar or superior to those in congenitally infected infants of women with primary CMV infection [6,7]. However, presently, laboratory or biological tests that predict cCMV infection in the fetuses of pregnant women with nonprimary CMV infection are not yet available.

Thus, screening methods that can predict cCMV infection on the basis of clinical information that is easily obtained in daily practice may be accessible and economically advantageous. Leruez-Ville et al. assessed the clinical risk factors of cCMV infection among both pregnant women with primary and nonprimary CMV infection in a large-scale study including 11,715 newborns who were screened by CMV-DNA PCR assays for saliva. They found that the clinical risk factors of cCMV infection among pregnant women with primary CMV infection were younger maternal age, parous, women born in high resources countries, and women from higher income groups. They also found that the clinical risk factors in pregnant women with nonprimary infection were younger maternal age and unemployment [35].

Our research group conducted two prospective cohort studies to determine the clinical factors predictive of cCMV infection in different populations.

The first prospective cohort study included 4125 pregnant women who delivered live-born infants in a primary maternity hospital between 2009 and 2017; for all infants born at the hospital, PCR was performed to analyze their urine for CMV DNA, and 9 newborns (0.22%) were diagnosed with cCMV infection (1 symptomatic, 8 asymptomatic) [36]. The clinical data were prospectively collected. Univariable and multivariable logistic regression analyses revealed that the presence of maternal fever/flu-like symptoms (odds ratio [OR], 17.9; 95% confidence interval [CI], 3.7–86.7; *p* < 0.001) and threatened miscarriage/premature labor in the second trimester (OR, 6.0; 95% CI, 1.6–22.8; *p* < 0.01) were clinical factors associated with cCMV infection among the women in the study who had normal or low-risk pregnancies. Notably, the combination of the presence of maternal fever/flu-like symptoms or threatened miscarriage/premature labor in the second trimester had 100% sensitivity, 53.2% specificity, 0.5% positive predictive value, and 100% negative predictive value (Youden index = 0.85). Additionally, the proportion of infants who had abnormal AABR test results was significantly higher among those with cCMV infection (11.1%) than among those without cCMV infection (0.5%; *p* < 0.05) [36].

The second prospective cohort study included 4380 pregnant women who delivered at a tertiary perinatal medical center between 2010 and 2019 [37]. cCMV infection was diagnosed in 32 infants (0.73%) by universal screening based on CMV-DNA PCR assays for newborns’ urine (20 symptomatic, 12 asymptomatic). The clinical data were prospectively collected. Univariable and multivariable logistic regression analyses revealed that a maternal age of <25 years (OR, 2.7; 95% CI, 1.1–6.6; *p* < 0.05), the presence of maternal fever/flu-like symptoms (OR, 5.4; 95% CI, 2.6–11.2; *p* < 0.01), fetal abnormalities found on ultrasonography (OR, 12.7; 95% CI, 5.8–27.7; *p* < 0.01), and preterm delivery at less than 34 gestational weeks (OR, 2.6; 95% CI, 1.1–6.0; *p* < 0.05) were independent clinical factors associated with cCMV infection among the women with high-risk pregnancies in this study. The combination of the presence of maternal fever/flu-like symptoms, fetal abnormalities found on ultrasonography, or preterm delivery at less than 34 gestational weeks had 90.6% sensitivity, 66.4% specificity, 1.9% positive predictive value, and 99.9% negative predictive value (Youden index = 0.57). Additionally, the proportion of low-birth weight infants, i.e., those infants whose birth weight was <2500 g, was significantly higher among the infants with cCMV infection (59.4%) than among those without cCMV infection (30.0%; *p* < 0.01) [37]. Here, we speculated that maternal fever/flu-like symptoms may have been caused by primary CMV infection, reinfection with a different strain of CMV, or reactivation of a latent CMV infection. Additionally, we proposed two hypotheses on the association between threatened miscarriage/premature labor or actual premature labor and the occurrence of cCMV infection: First, intrauterine CMV infection may itself cause threatened miscarriage/premature delivery. Second, inflammatory conditions underlying threatened miscarriage/premature delivery might induce differentiation of latently CMV infected monocytes, and CMV infection might be reactivated and transmitted to a fetus via the placenta [38].

The biological and clinical factors associated with increased risk of cCMV infection are summarized in Table 1 [26,28,29,34,35,36,37].

## 6. An Example of Potential Screening and Therapeutic Strategies for Improving Outcomes of Infants with Symptomatic cCMV Infection

At present, neither maternal nor neonatal universal screening for cCMV infection are recommended because there are no established fetal and neonatal therapies for affected fetuses and infants. However, as described previously, it has been recently demonstrated that early antiviral therapies with oral valganciclovir may improve neurological outcomes in newborns with symptomatic cCMV infection [2,3,4]. In addition, some previous clinical trials showed the effectiveness of fetal therapies for symptomatic cCMV infection, e.g., hyper-immunoglobulin injection into maternal blood [39] or into the fetal peritoneal cavity [40], or oral administration of high-dosage valacyclovir to mothers [41], etc. Furthermore, a recent clinical trial showed that the combination of fetal therapy with immunoglobulin and of neonatal therapies with antiviral drugs might be more effective in improving neurological outcomes in newborns with symptomatic cCMV infection as compared to neonatal therapy alone [42].

Neither established neonatal nor fetal therapies for cCMV infection were present. An example of a flow algorithm of potential screening and treatment for cCMV infection based on our recent research findings is shown in Figure 1. Because a larger number of infants with cCMV infection are born to mothers with non-primary CMV infection than those with primary infection, maternal serological CMV screening by CMV IgG/IgM or CMV IgG avidity measurements for identifying pregnant women who have primary CMV infection during pregnancy may be unnecessary. In addition, because the risk assessment of cCMV infection based on complex biological assays, including epitope-specific antibody detection assays (described in the Section 3.2) and CMV ELISpot assays (described in the Section 4), are not commonly available, we made a flow algorithm which was mainly comprised of the clinical risk factors of cCMV infection. Not only CMV-seronegative pregnant women but also seropositive ones may have to receive educational intervention to prevent primary CMV infection or reinfection with different strains of CMV during pregnancy. Fetal therapies might be considered against fetuses who are prenatally diagnosed with symptomatic cCMV infection based on positive CMV DNA PCR results in the amniotic fluid and on ultrasound fetal abnormalities associated with cCMV infection, if the pregnant woman in question and her family desire. On the other hand, all infants who are born to mothers who have risk factors of cCMV infection, including fever of flu-like symptoms, fetal abnormalities found on ultrasonography, threatened miscarriage/premature labor, preterm labor at less than 34 gestational weeks, and all infants with risk factors, including low birth weight and abnormal AABR test results, should probably undergo CMV DNA PCR analysis in newborn urine to confirm the presence of cCMV infection. Newborns with positive PCR results in their urine should be worked up to identify the symptoms of cCMV infection, including physical and neurological examinations, cerebral ultrasound, auditory brain-stem response test, ophthalmoscopy, and head computed tomography/magnetic resonance imaging. Neonatal treatment with valganciclovir against symptomatic cCMV infection might be considered against infants with symptomatic cCMV infection. However, further investigations are needed to estimate the utility of our strategies shown in Figure 1, and also to establish implementable screening and therapeutic strategies for cCMV infection.

Methods for screening or diagnosis of congenital CMV infection are indicated in gray. A final diagnosis of congenital CMV infection is indicated in red. Interventions for treating symptomatic congenital CMV infection are indicated in blue. Abbreviations: CMV, cytomegalovirus; GW, gestational weeks; IgG, immunoglobulin G; NAbs, neutralizing antibodies; PCR, polymerase chain reaction; AABR, automated auditory brainstem response; ABR, auditory brainstem response; CT, computed tomography; MRI, magnetic resonance imaging.

## 7. Approved Antiviral Drugs and Candidate Vaccines against HCMV

The development of effective and safe antiviral drugs and vaccines against HCMV has been desired.

At present, five anti-HCMV drugs have been approved. These antiviral drugs are categorized into 2 groups: (1) drugs inhibiting the synthesis of viral DNA by targeting the viral DNA polymerase, including ganciclovir (GCV), valganciclovir (VGCV), cidofovir (CDV), and foscarnet (FOS); (2) drugs inhibiting the packaging of viral DNA into capsids by terminase complex, including letermovir (LTV). VGCV is a prodrug of GCV, and GCV and VGCV were approved for the treatment of HCMV diseases in immunocompromised hosts or infants with cCMV infection, and for the prophylaxis of HCMV diseases. CDV was approved for the treatment of HCMV retinitis in patients infected with human immunodeficiency virus. FOS was approved for the treatment of HCMV diseases, and this drug is especially useful for the treatment of GCV-resistant HCMV infection. LTV is a new class of approved drugs for the prophylaxis of HCMV infection in allogenic stem cell transplant recipients. As mentioned above, at present, antiviral drugs that were reported to be safe and effective for treatment of infants with symptomatic cCMV infection are GCV and VGCV only [43].

The main candidate vaccines developed up to now are as follows: (1) live-attenuated vaccines, e.g., V160, which is composed by an attenuated AD169 strain where the PC expression is restored; (2) subunit vaccines, e.g., gB/MF59, which is generated by combing gB with an adjuvant (MF59); (3) virus vectored vaccines, e.g., Triplex, which is an attenuated poxvirus modified Vaccinia Ankara encoding pp65, IE1-exon4, and IE2-exon5; (4) chimeric peptidic vaccines, e.g., CMVPepVax, which is a pp65 fused to either pan DR helper T lymphocyte epitope or natural tetanus sequence; (5) vaccine based on enveloped virus-like particles, e.g.,VBI-1501, which expresses the extracellular domain of gB fused with the transmembrane and cytoplasmic domains from vesicular stomatitis virus G protein; (6) plasmid-based DNA vaccines, e.g., VCL-CB01, which is a bivalent HCMV DNA vaccine that contains two plasmids, VCL-6368 and VCL-6365, encoding pp65 and gB; (7) RNA-based vaccines, e.g., mRNA-1647, which is composed of 1 mRNA encoding gB and 5 mRNAs encoding PC; and (8) peptide vaccines, e.g., a CMV pp65 peptide vaccine. Regarding clinical trials of vaccines enrolling women of childbearing age, phase 2 clinical trials of live-attenuated vaccines (V160), subunit vaccines (gB/MF59), and RNA-based vaccines (mRNA-1647) were conducted. In the phase 2 clinical trials of gB/MF59, vaccine efficacies were reported to be 43–53%. Meanwhile, the results of the phase 2 clinical trials of V160 and mRNA-1647 are not yet available [44].

## 8. Conclusions

Early antiviral therapies for newborns with symptomatic cCMV infection may improve neurological outcomes in affected children [2,3,4]. Conversely, at present, there is no evidence of the benefit of antiviral therapies for infants with asymptomatic cCMV infection. Furthermore, antiviral therapies for newborns with mildly symptomatic cCMV infection or affected newborns more than 30 days old are not recommend because of the lack of evidence [45]. However, Andrea et al. demonstrated that 56% (19/34) of newborns who seemed to have asymptomatic cCMV infection by normal physical examinations had abnormalities on laboratory or imaging examinations [46]. Therefore, newborns at high risk of cCMV infection, including cases who have no obvious morphological abnormalities associated with cCMV infection, should be identified.

Nevertheless, both primary CMV infection and nonprimary CMV infection during pregnancy, including reinfection and reactivation, can cause severe symptomatic cCMV infection in the infant. Understanding host immunity against CMV and how CMV evades host immunity is important both for elucidating the mechanism whereby both primary and nonprimary CMV infection can cause cCMV infection and for developing preventive HCMV vaccines.

Previous investigators developed ELISA methods for detecting epitope-specific antibodies against the pentameric complex on HCMV or against epitopes on several different strains of HCMV, finding associations between the quantities of these antibodies and susceptibility to cCMV infection. Other investigators have suggested that CMV-specific T cell-mediated immune responses measured by CMV ELISpot assay may be useful for predicting cCMV infection. However, these assays are nonstandardized and not commonly available in daily practice, and interpretation of the data is often complex.

Consequently, screening for cCMV infection that is based on clinical risk factors that are easily obtained in daily practice—such as the presence of maternal fever/flu-like symptoms, threatened miscarriage/premature labor, fetal abnormalities found on ultrasonography that are associated with cCMV infection, premature deliveries, low birth weight, or abnormal AABR test results—may be clinically and economically advantageous. Our previous studies, including a total of 21 symptomatic and 20 asymptomatic infected newborns, implied that screening strategies for cCMV infection based on clinical risk factors could identify not only symptomatic newborns but also asymptomatic ones with a high sensitivity of 92.7% (38/41) [36,37]. A prospective cohort study to assess the utility of such screening strategies is now ongoing.

## Figures and Tables

**Figure 1 ijms-22-13487-f001:**
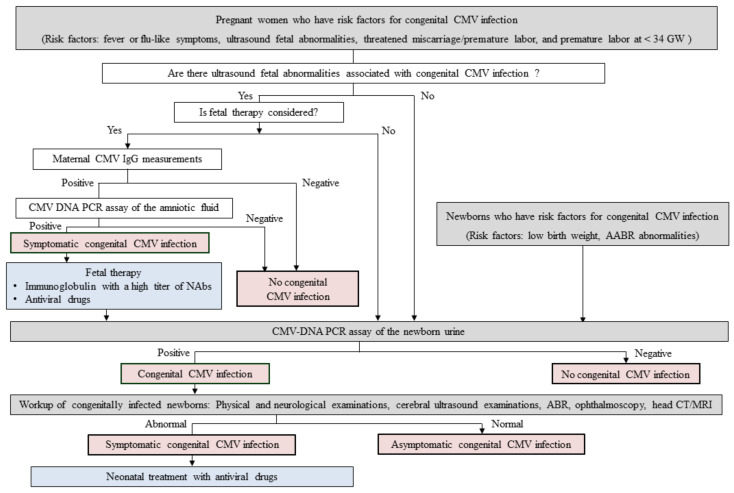
An example of a flow algorithm of potential screening and treatment for congenital CMV infection based on our recent researches.

**Table 1 ijms-22-13487-t001:** Summary of biological and clinical factors associated with increased risk of congenital CMV infection.

Factors	References
Biological factors	
A low CMV-specific IgG avidity index	Sonoyama et al., 2012 [26]
A delay in the production of antibodies against pentameric complex during primary infection	Lilleri et al., 2013 [28]
An absence of antibodies against pentameric complex and a low CMV-specific IgG avidity index	Kaneko et al., 2017 [29]
High CMV levels on ELISpot, viremia/viruria, and low CMV IgG avidity index	Forner et al., 2016 [34]
Clinical factors	
Younger age and multiparity in high-resource countries after primary infection Higher incomes after primary infection Younger age and unemployment after nonprimary infection	Leruez-Ville et al., 2017 [35]
Fever/flu-like symptoms and threatened miscarriage/premature labor in the second trimester in low-risk populations	Uchida et al., 2020 [36]
Younger age, fever/flu-like symptoms, fetal ultrasound abnormalities, and preterm delivery at <34 gestational weeks in high-risk populations	Imafuku et al., 2020 [37]

Abbreviations: CMV, cytomegalovirus; ELISpot, enzyme-linked immunosorbent spot; IgG, immunoglobulin G.

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
