# Peer review of "The Current Challenges in Developing Biological and Clinical Predictors of Congenital Cytomegalovirus Infection"

_ijms, 2021, doi:10.3390/ijms222413487_

Round 1

Reviewer 1 Report

Summary

The review article „Promising biological and clinical predictors of congenital cytomegalovirus infection” by Tanimura et al covers an interesting and important topic. The authors discuss immune responses to human cytomegalovirus infection and diagnostic tests of these responses. They then move onto clinical indicators, and sketch out a potential screening strategy to identify affected foetuses and neonates.

General Comments

Although the topic is worthy of review, this paper unfortunately requires a clearer structure and firmer conclusions to be of real use to the reader. For instance, section 2, Host Immune responses to HCMV does not completely describe either the host’s defences against the virus or the virus’s immune evasion and modulation mechanisms. Although this subject is large and complex, an overview would be possible, perhaps with indications as to which areas are most relevant to this review, such as those aspects of the immune response that feature in the diagnostic tests described later. Section 2 is generally not well linked to the rest of the manuscript.

The next section discusses serological assays. Again, this section is a mixture of some of what is known about antibody responses to HCMV and then some individual studies relating to relating to the use of antibodies in predicting cCMV. This section could be better structured for clarity, with the information on antibody responses fitting better in section 2. Section 4 then describes T cell assays. The assays described in section 3 and 4 are then not further discussed, the rest of the paper focuses on clinical factors, largely drawing on the authors’ own previously published studies. The results of these studies are very interesting, but make up too much of the second half of the paper, with not enough balance from other researchers. Finally, the authors propose a screening system for symptomatic cCMV. While possible, as the authors themselves state, without defined and approved treatment options there is no reason to implement a general screening programme. The need for this screening needs to be more carefully explained, particularly with potential treatment options discussed in more detail, to be convincing. The authors imply that treatment options are already available and approved, although this is not the case. The suggestions for future screening also do not incorporate any of the biological assays, which may be a valid viewpoint, but should be explained, as the paper otherwise consists of unlinked sections.

The authors make the point well that pre-existing maternal seropositivity does not protect against cCMV, which is valid and needs to be more generally known.

The authors have obvious expertise in the area and their own studies that are described here are highly relevant. It therefore seems likely that a reworking of this review would result in a much more successful manuscript.

Specific Comments

Abstract: it is implied that early antiviral therapy for symptomatic cCMV is approved and effective, which is not yet the case. It is also implied that maternal screening for cCMV is routinely performed by serological screening, which is also incorrect.

Introduction: Lines 51 to 53: maternal serological tests are referred to as it they were used to diagnose cCMV. This is not correct. Maternal serology may often be used incorrectly to predict risk of cCMV development, with the belief that HCMV seropositivity implies protection, but not as a tool to diagnose cCMV. The point that cCMV occurs more often when maternal seropositivity is higher is, however an important one. This was also discussed by William Britt in 2015 in Med Microbiol Immunol 204:263-271.

Line 72: Innate Immune Response: this should be changed to Innate Immune cells, otherwise innate immunity eg pattern recognition could be expected.

It should then be clarified which innate immune cells are infected by HCMV and which are not, but are affected by HCMV, such as NK cells. The effects of HCMV on NK cells is a large topic, which should at least be referred to.

Line 94 onwards: CD4 T cells can have cytotoxic functions in HCMV infections. These cells have also been identified in humans and are important.

Line 98 onwards: B and T cell responses are confusingly mixed.

Line 123: It is not universally agreed that pre-existing maternal immunity reduces the risk of cCMV infection, and the figure of 65% is not generally known. See again Britt, 2015.

Line 146: This statement is incorrect. Maternal reinfection was shown to increase the risk of cCMV, but not in comparison to maternal reactivation. It is unknown how often reactivation occurs and it was not shown in the cited manuscript.

Lines 271-281: While some studies show benefit of therapies, they are not yet in general use and their benefit is often marginal. It should be made clear that there is still an important unmet clinical need, and until therapies for cCMV infected foetuses and neonates are improved, screening is not yet implementable.

Lines 287-288: The sentence should read “ not only CMV-seronegative pregnant women but also seropositive ones….”

Table 1: In discussing the screening strategy, the biological predictors discussed earlier should be referred to. If the authors wish to state that only clinical parameters should be used, this viewpoint should be defended.

Lines 314 onwards: The screening is designed to detect symptomatic foetuses and newborns. Is there evidence that symptomatic infections are often missed? Screening seems more likely to detect otherwise overlooked asymptomatic infections, which do not fall into the treatment plan suggested by the authors. Can evidence be shown that this screening would result in more symptomatic cases being identified? The conclusions mix a discussion of how to identify all cases of cCMV and the proposed therapeutically beneficial screening for symptomatic cases. Is it the authors’ opinion that all cCMV cases should be identified? There may well be good reason to do this, but if so this should be explained.

Author Response

Point-by-point response to reviewers’ comments

Reviewer #1:

Overall Comment: The review article “Promising biological and clinical predictors of congenital cytomegalovirus infection” by Tanimura et al covers an interesting and important topic. The authors discuss immune responses to human cytomegalovirus infection and diagnostic tests of these responses. They then move onto clinical indicators, and sketch out a potential screening strategy to identify affected foetuses and neonates.

Response: We thank you very much for positive evaluating our review and providing valuable comments.

General Comments:

Comment 1: Although the topic is worthy of review, this paper unfortunately requires a clearer structure and firmer conclusions to be of real use to the reader. For instance, section 2, Host Immune responses to HCMV does not completely describe either the host’s defences against the virus or the virus’s immune evasion and modulation mechanisms. Although this subject is large and complex, an overview would be possible, perhaps with indications as to which areas are most relevant to this review, such as those aspects of the immune response that feature in the diagnostic tests described later. Section 2 is generally not well linked to the rest of the manuscript.                                  

Response to Comment 1: As reviewer pointed out, Section 2.1 is not linked to the rest of the manuscript at all. But we think that the sentences regarding the immune response of CD4+/CD8+T cells in Section 2.2 were linked to contents of Section 4 (Assays for Measuring CMV-Specific T Cell-Mediated Immunity), and that the sentences regarding the production of neutralizing antibodies with high avidity to HCMV in Section 2.2 were also linked to the contents of Section 3.2. (epitope-specific antibody). According to the reviewer’s suggestion, the sentence in Lines 125–128 was revised to “Thus, several researchers have attempted to investigate predictors of cCMV infection by measuring neutralizing antibodies (this issue is described in Section 3.2.) or by evaluating the magnitude of CD4+ and CD8+ T cell responses (this issue is described in Section 4).” were

Comment 2:

Comment 2-1: The next section discusses serological assays. Again, this section is a mixture of some of what is known about antibody responses to HCMV and then some individual studies relating to relating to the use of antibodies in predicting cCMV. This section could be better structured for clarity, with the information on antibody responses fitting better in section 2.

Response to Comment 2-1: According to the reviewer’s suggestion, in Line 182 and Line 217, “, as described in Section 2.2.” and “as described in Section 2.2” were added, respectively.

Comment 2-2: Section 4 then describes T cell assays. The assays described in section 3 and 4 are then not further discussed, the rest of the paper focuses on clinical factors, largely drawing on the authors’ own previously published studies. The results of these studies are very interesting, but make up too much of the second half of the paper, with not enough balance from other researchers.

Response to Comment 2-2: According to the reviewer’s suggestion, in Lines 245–252, the results of another researcher’s study were added, and we shorten the sentences regarding the results of our previous studies in Section 5.

We added the reference below.

  1. Leruez-Ville, et al. Risk Factors for Congenital Cytomegalovirus Infection Following Primary and Nonprimary Maternal Infection: A Prospective Neonatal Screening Study Using Polymerase Chain Reaction in Saliva. Clin Infect Dis. 2017,65,398-404.

Comment 2-3: Finally, the authors propose a screening system for symptomatic cCMV. While possible, as the authors themselves state, without defined and approved treatment options there is no reason to implement a general screening programme. The need for this screening needs to be more carefully explained, particularly with potential treatment options discussed in more detail, to be convincing. The authors imply that treatment options are already available and approved, although this is not the case. The suggestions for future screening also do not incorporate any of the biological assays, which may be a valid viewpoint, but should be explained, as the paper otherwise consists of unlinked sections.

Response to Comment 2-3: As the reviewer pointed out, there are no established treatment for fetuses or infants with cCMV infection. However, previous clinical trials demonstrated that fetal therapies with immunoglobulin and neonatal therapies with antiviral drugs may improve neurological outcomes in affected children. In addition, epitope-specific antibody detection assays and CMV ELISpot assays are not commonly available at present. According to the reviewer’s suggestion, we revised the title of Section 6 and some sentences in the Section. Especially, we soften the expressions regarding the utility of our screening method and the effectiveness of fetal and neonatal therapies in Section 6.

Comment 3: The authors make the point well that pre-existing maternal seropositivity does not protect against cCMV, which is valid and needs to be more generally known. The authors have obvious expertise in the area and their own studies that are described here are highly relevant. It therefore seems likely that a reworking of this review would result in a much more successful manuscript.

Response: We thank you very much for positive evaluating our review and providing valuable comments.

Specific Comments:

Comment 4: Abstract: it is implied that early antiviral therapy for symptomatic cCMV is approved and effective, which is not yet the case. It is also implied that maternal screening for cCMV is routinely performed by serological screening, which is also incorrect.

Response to Comment 4: According to reviewer’s suggestion, sentences in Lines 17–22 in the Abstract were revised to “Recent studies have demonstrated that early antiviral therapy for infants with symptomatic congenital CMV (cCMV) infection may improve neurological outcomes; thus, accurate identification of newborns at high risk for cCMV infection may contribute to improve outcomes in affected children. However, maternal serological screening for cCMV infection by diagnosing primary infection during pregnancy, which is one of the popular screening strategies, …..”.

Comment 5: Introduction: Lines 51 to 53: maternal serological tests are referred to as it they were used to diagnose cCMV. This is not correct. Maternal serology may often be used incorrectly to predict risk of cCMV development, with the belief that HCMV seropositivity implies protection, but not as a tool to diagnose cCMV. The point that cCMV occurs more often when maternal seropositivity is higher is, however an important one. This was also discussed by William Britt in 2015 in Med Microbiol Immunol 204:263-271.

Response to Comment 5: According to reviewer’s suggestion, we added the reference.

  1. Britt, W. Controversies in the natural history of congenital human cytomegalovirus infection: the paradox of infection and disease in offspring of women with immunity prior to pregnancy. Med Microbiol Immunol. 2015,204,263-271.

And Lines 54–61 in the Introduction, “Therefore, serological tests, including maternal blood tests of CMV-specific immunoglobulin G (IgG) and CMV-specific immunoglobulin M (IgM), are commonly used in practice because it is conventionally thought that preexisting maternal human CMV (HCMV) immunity can exert the protective effect against intrauterine transmission and, therefore, that almost all symptomatic cCMV infections in infants are caused by maternal primary CMV infection either during or just before pregnancy. Recent studies, however, have demonstrated that preexisting maternal HCMV immunity provides only limited protection against transmission to the fetuses [5], and also that …..”

were added or revised.

Comment 6: Line 72: Innate Immune Response: this should be changed to Innate Immune cells, otherwise innate immunity eg pattern recognition could be expected.

Response to Comment 6: According to reviewer’s suggestion, Line 76 was changed to “2.1. Innate Immune Cells”.

Comment 7: It should then be clarified which innate immune cells are infected by HCMV and which are not, but are affected by HCMV, such as NK cells. The effects of HCMV on NK cells is a large topic, which should at least be referred to.

Response to Comment 7: According to reviewer’s suggestion, in Lines 90–92, “HCMV can infect polymorphonuclear leukocytes and monocytes, conversely, the virus cannot infect NK cells but can affected their function.” was added.

And, in Lines 92–101, “A balance of signals generated by activating and inhibitory receptors decides whether NK cells do or do not kill the infected cells. NKG2D is one of the activating NK cell receptors in human, and its ligands include MHC class I chain-related protein A (MICA), MHC class I chain-related protein B (MICB) and UL 16-binding protein 1 (ULBP1) to ULBP6. Expression of these ligands on the infected cells induces NK cell cytotoxicity. In contrast, the UL 16 and UL 142 glycoprotein of HCMV downregulate ULBPs, MICA, and MICB by causing intracellular retention of these ligand proteins [13-15]. On the other hand, US 18 and US 20 downregulate MICA by targeting the ligand protein for lysosomal degradation [16]. Thus, HCMV evades NK cell attack by downregulation of ligands of activating NK cell receptors.” was added.

And, we added the references below.

  1. Cosman, D.;Mullberg, J.;Sutherland, C.L.;Chin, W.;Armitage, R.;Fanslow, W.;Kubin, M.; Chalupny, N.J. ULBPs, novel MHC class I-related molecules, bind to CMV glycoprotein UL16 and stimulate NK cytotoxicity through the NKG2D receptor. Immunity. 2001,14,123-133.
  2. Wu, J.;Chalupny, N.J.;Manley, T.J.;Riddell, S.R.;Cosman, D.; Spies, T. Intracellular retention of the MHC class I-related chain B ligand of NKG2D by the human cytomegalovirus UL16 glycoprotein. J Immunol. 2003,170,4196-4200.
  3. Chalupny, N.J.;Rein-Weston, A.;Dosch, S.; Cosman, D. Down-regulation of the NKG2D ligand MICA by the human cytomegalovirus glycoprotein UL142. Biochem Biophys Res Commun. 2006,346,175-181.
  4. Fielding, C.A.;Aicheler, R.;Stanton, R.J.;Wang, E.C.;Han, S.;Seirafian, S.;Davies, J.;McSharry, B.P.;Weekes, M.P.;Antrobus, P.R.; et al. Two novel human cytomegalovirus NK cell evasion functions target MICA for lysosomal degradation. PLoS Pathog. 2014,10,e1004058.

Comment 8: Line 94 onwards: CD4 T cells can have cytotoxic functions in HCMV infections. These cells have also been identified in humans and are important.

Response to Comment 8: According to reviewer’s suggestion, in Lines 111–114, “In addition, it has been recently shown that CD4+ T cells exhibit antiviral activity, which is independent of their helper function, through production of cytokines IFN-g and TNF, and through direct cytolytic actions via perforin-dependent and Fas-dependent killing [18].” was added.

And, we added the reference.

  1. Juno, J.A.;van Bockel, D.;Kent, S.J.;Kelleher, A.D.;Zaunders, J.J.; Munier, C.M. Cytotoxic CD4 T Cells-Friend or Foe during Viral Infection? Front Immunol. 2017,8,19.

Comment 9: Line 98 onwards: B and T cell responses are confusingly mixed.

Response to Comment 9: According to reviewer’s suggestion, the sentence in Lines 116–118 was revised to “Previous studies demonstrated that maternal neutralizing antibodies with high avidity to HCMV, which are produced by antigen-specific B cells activated by helper CD4+ T cells, can decrease the risk of cCMV infection [20]”

Comment 10:  Line 123: It is not universally agreed that pre-existing maternal immunity reduces the risk of cCMV infection, and the figure of 65% is not generally known. See again Britt, 2015.

Response to Comment 10: According to reviewer’s suggestion, the sentence in Lines 168–169 was revised to “Nevertheless, preexisting maternal HCMV immunity provides only limited protection against intrauterine HCMV transmission [5].”

Comment 11:  Line 146: This statement is incorrect. Maternal reinfection was shown to increase the risk of cCMV, but not in comparison to maternal reactivation. It is unknown how often reactivation occurs and it was not shown in the cited manuscript.

Response to Comment 11: According to reviewer’s suggestion, the sentence in Lines 190–192 was revised to “Conversely, the results of previous studies suggested that the fetuses of pregnant women who were reinfected with different strains of HCMV were at higher risk for cCMV infection [30].”

Comment 12:  Lines 271-281: While some studies show benefit of therapies, they are not yet in general use and their benefit is often marginal. It should be made clear that there is still an important unmet clinical need, and until therapies for cCMV infected foetuses and neonates are improved, screening is not yet implementable.

Response to Comment 12: According to reviewer’s suggestion, the title of the Section 6 was changed to “6. An example of potential screening and therapeutic strategies for improving outcomes of infants with symptomatic cCMV infection”.

And, the sentence in Lines 310–311 was revised to “….. early antiviral therapies with oral valganciclovir may improve neurological outcomes in newborns with symptomatic cCMV infection [2-4].”

And, the sentences in Lines 319–321 were revised to “Although, neither established neonatal nor fetal therapies for cCMV infection were present. An example of a flow algorithm of potential screening and treatment for cCMV infection based on our recent research findings is shown in Figure 1”.

And, in Lines 346–348, “However, further investigations are needed to estimate the utility of our strategies shown in Figure 1, and also to establish implementable screening and therapeutic strategies for cCMV infection.” was added.

And, the title of Figure 1 was changed to “An example of a flow algorithm of potential screening and treatment for congenital CMV infection based on our recent researches.”

Comment 13: Lines 287-288: The sentence should read “not only CMV-seronegative pregnant women but also seropositive ones….”

Response to Comment 13: According to reviewer’s suggestion, the sentence in Line 330 was revised to “Not only CMV-seronegative pregnant women but also seropositive ones may have to….”

Comment 14: Table 1: In discussing the screening strategy, the biological predictors discussed earlier should be referred to. If the authors wish to state that only clinical parameters should be used, this viewpoint should be defended.

Response to Comment 14: According to reviewer’s suggestion, all references, which were not included in the main text, were removed from the revised Table1.

And, the sentence in Lines 164–166 was revised to “Because the CMV-specific IgG avidity index increases over time, a low CMV-specific IgG avidity index, which indicates a recent CMV infection, was reported to be a significant predictor of cCMV infection [26]”, and we added the reference below.

  1. Sonoyama, A.;Ebina, Y.;Morioka, I.;Tanimura, K.;Morizane, M.;Tairaku, S.;Minematsu, T.;Inoue, N.; Yamada, H. Low IgG avidity and ultrasound fetal abnormality predict congenital cytomegalovirus infection. J Med Virol. 2012,84,1928-1933.

And, in Lines 245–252, “Leruez-Ville et al. assessed the clinical risk factors of cCMV infection among both pregnant women with primary and nonprimary CMV infection in large-scale study including 11715 newborns who were screened by CMV-DNA PCR assays for saliva. And they found that the clinical risk factors of cCMV infection among pregnant women with primary CMV infection were younger maternal age, parous, women born in high resources countries, and women from higher income groups. They also found that the clinical risk factors in pregnant women with nonprimary infection were younger maternal age and unemployed [35].” were added.

And, in Lines 323–329, “….. maternal serological CMV screening by CMV IgG/IgM or CMV IgG avidity measurements for identifying pregnant women who have primary CMV infection during pregnancy may be unnecessary. In addition, because the risk assessment of cCMV infection based on complex biological assays, including epitope-specific antibody detection assays (described in the section 3.2) and CMV ELISpot assays (described in the section 4), are not commonly available, we made such a flow algorithm which was mainly comprised of clinical risk factors of cCMV infection.” were revised and added.

Comment 15: Lines 314 onwards: The screening is designed to detect symptomatic foetuses and newborns. Is there evidence that symptomatic infections are often missed? Screening seems more likely to detect otherwise overlooked asymptomatic infections, which do not fall into the treatment plan suggested by the authors. Can evidence be shown that this screening would result in more symptomatic cases being identified? The conclusions mix a discussion of how to identify all cases of cCMV and the proposed therapeutically beneficial screening for symptomatic cases. Is it the authors’ opinion that all cCMV cases should be identified? There may well be good reason to do this, but if so this should be explained.

Response to Comment 15: According to reviewer’s suggestion, in Lines395–403, “Conversely, at present, there is no evidence of the benefit of antiviral therapies for infants with asymptomatic cCMV infection. Furthermore, antiviral therapies for newborns with mildly symptomatic cCMV infection or affected ones more than 30 days old are not recommend because of the lack of evidence [45]. However, Andrea et al. demonstrated that 56% (19/34) of newborns, who seemed to have asymptomatic cCMV infection by normal physical examinations, had abnormalities on laboratory or imaging examinations [46]. Therefore, newborns at high risk for cCMV infection, including cases who have no obvious morphological abnormalities associated with cCMV infection, should be identified.” were added.

And, in Lines 422–425, “Our previous studies, including total 21 symptomatic and 20 asymptomatic infected newborns, implied that screening strategies for cCMV infection based on clinical risk factors could identify not only symptomatic newborns but also asymptomatic ones with high sensitivity of 92.7% (38/41) [36,37].” was added.

And, we added the references below.

  1. Chiopris, G.;Veronese, P.;Cusenza, F.;Procaccianti, M.;Perrone, S.;Dacco, V.;Colombo, C.; Esposito, S. Congenital Cytomegalovirus Infection: Update on Diagnosis and Treatment. Microorganisms. 2020,8.
  2. Ronchi, A.;Zeray, F.;Lee, L.E.;Owen, K.E.;Shoup, A.G.;Garcia, F.;Vazquez, L.N.;Cantey, J.B.;Varghese, S.;Pugni, L.; et al. Evaluation of clinically asymptomatic high risk infants with congenital cytomegalovirus infection. J Perinatol. 2020,40,89-96.

Reviewer 2 Report

The current review of literature on cCMV infection is very relevant and is written well. It is largely free of typographical/grammar mistakes. I have a few minor comments about the manuscript.

The title " Promising biological and clinical predictors of congenital cytomegalovirus infection" seems to promise more than what is being offered in the review so authors may want to revise it. Briefly, the review discusses the current challenges and available measures to diagnose and predict the severity of cCMV infection; however, it does not offer any promising predictors that are not already known to the field. The presence of maternal fever/flu-like symptoms may be correlated with cCMV but this is still speculation, thus the flow chart (Fig 1) where the decision tree relies on these risk factors is flawed.

I believe the 'host immune responses to HCMV' section is very basic and does not cover the vast literature available to us on this topic. Authors should at least talk about viral proteins and how they downregulate MHC. They should also cover CMV latency, reactivation, and spread.

Although authors list the challenges in current diagnostics, they do not offer a solution.

In the "Potential screening and therapeutic strategies for cCMV infection", the authors do not include some of the recently approved antivirals as well as vaccines that are in an advanced stage of clinical trials.

Author Response

Point-by-point response to reviewers’ comments

Reviewer #2:

Overall Comment: The current review of literature on cCMV infection is very relevant and is written well. It is largely free of typographical/grammar mistakes. I have a few minor comments about the manuscript.

Response: We thank you very much for positive evaluating our review and providing valuable comments.

Specific Comments:

Comment 1: The title " Promising biological and clinical predictors of congenital cytomegalovirus infection" seems to promise more than what is being offered in the review so authors may want to revise it. Briefly, the review discusses the current challenges and available measures to diagnose and predict the severity of cCMV infection; however, it does not offer any promising predictors that are not already known to the field. The presence of maternal fever/flu-like symptoms may be correlated with cCMV but this is still speculation, thus the flow chart (Fig 1) where the decision tree relies on these risk factors is flawed.

Response to Comment 1: According to reviewer’s suggestion, we changed the title to “The current challenges in developing biological and clinical predictors of congenital cytomegalovirus infection”.

As the reviewer pointed out, our screening and therapeutic strategy shown in Figure 1 was not established at present. But, we want to propose our screening approach for predicting cCMV infection as one of the potential screening methods. Our screening approach is mainly composed of clinical factors, which are easily obtained in daily practice by clinical interviews or ultrasound examinations, therefore our screening approach may be clinically and economically advantageous. According to reviewer’s suggestion, I weakened our claim and revised some sentences. For example, the figure legend was changed to “Figure 1. An example of a flow algorithm of potential screening and treatment for congenital CMV infection based on our recent researches.”, and in Lines 346–348, “However, further investigations are needed to estimate the utility of our strategies shown in Figure 1, and also to establish implementable screening and therapeutic strategies for cCMV infection.” was added.

Comment 2: I believe the 'host immune responses to HCMV' section is very basic and does not cover the vast literature available to us on this topic. Authors should at least talk about viral proteins and how they downregulate MHC. They should also cover CMV latency, reactivation, and spread.

Response to Comment 2: According to reviewer’s suggestion, in Lines 83–87, “Concretely, during the early phase of viral replication, HCMV gene product U3 inhibits intracellular transport of MHC class I molecules toward the cell surface. US2 and US11 dislocate newly generated MHC class I molecules from the endoplasmic reticulum (ER) to the cytoplasm, where they are degraded by the proteasome [11].” was added.

And, we added the reference.

  1. Gabor, F.;Jahn, G.;Sedmak, D.D.; Sinzger, C. In vivo Downregulation of MHC Class I Molecules by HCMV Occurs During All Phases of Viral Replication but Is Not Always Complete. Front Cell Infect Microbiol. 2020,10,283.

According to reviewer’s suggestion, in Lines 129–153, a new Section “2.3. The Establishment of Latency and Reactivation of HCMV” was added. And, the title of the Section 2 was changed to “2. The Struggle between HCMV and Host Immunity”.

And, we added the reference.

  1. Forte, E.;Zhang, Z.;Thorp, E.B.; Hummel, M. Cytomegalovirus Latency and Reactivation: An Intricate Interplay With the Host Immune Response. Front Cell Infect Microbiol. 2020,10,130.

Comment 3: Although authors list the challenges in current diagnostics, they do not offer a solution.

Response to Comment 3: Unfortunately, we cannot offer any solutions at present, because there are no evidence-based guidelines for screening and treating infants with cCMV infection. The fact is a major limitation of the field of perinatology. Therefore, in Lines 346–348, “However, further investigations are needed to estimate the utility of our strategies shown in Figure 1, and also to establish implementable screening and therapeutic strategies for cCMV infection.” was added as a limitation in our review.

Comment 4: In the "Potential screening and therapeutic strategies for cCMV infection", the authors do not include some of the recently approved antivirals as well as vaccines that are in an advanced stage of clinical trials.

Response to Comment 4: According to reviewer’s suggestion, in Lines 358–392, a new Section “7. Approved Antiviral drugs and Candidate Vaccines against HCMV” was added.

And, we added the reference.

  1. Britt, W.J.; Prichard, M.N. New therapies for human cytomegalovirus infections. Antiviral Res. 2018,159,153-174.
  2. Scarpini, S.;Morigi, F.;Betti, L.;Dondi, A.;Biagi, C.; Lanari, M. Development of a Vaccine against Human Cytomegalovirus: Advances, Barriers, and Implications for the Clinical Practice. Vaccines (Basel). 2021,9.

Round 2

Reviewer 1 Report

The authors have addressed the problems referred to in the previous review and greatly improved the manuscript.

Although the manuscript could benefit from further language editing, it is understandable.

Reviewer 2 Report

The manuscript has been revised to tone down some of the claims made earlier and several relevant sections have been added. It is now well suited for publication.